# The Effect of Oxidative Stress-Induced Autophagy by Cadmium Exposure in Kidney, Liver, and Bone Damage, and Neurotoxicity

**DOI:** 10.3390/ijms232113491

**Published:** 2022-11-04

**Authors:** Yonggang Ma, Qunchao Su, Chengguang Yue, Hui Zou, Jiaqiao Zhu, Hongyan Zhao, Ruilong Song, Zongping Liu

**Affiliations:** 1College of Veterinary Medicine, Yangzhou University, Yangzhou 225009, China; 2Jiangsu Co-Innovation Center for Prevention and Control of Important Animal Infectious Diseases and Zoonoses, Yangzhou 225009, China; 3Joint International Research Laboratory of Agriculture and Agri-Product Safety, Ministry of Education of China, Yangzhou University, Yangzhou 225009, China

**Keywords:** cadmium, autophagy, apoptosis, pyroptosis, ferroptosis, oxidative stress

## Abstract

Environmental and occupational exposure to cadmium has been shown to induce kidney damage, liver injury, neurodegenerative disease, and osteoporosis. However, the mechanism by which cadmium induces autophagy in these diseases remains unclear. Studies have shown that cadmium is an effective inducer of oxidative stress, DNA damage, ER stress, and autophagy, which are thought to be adaptive stress responses that allow cells exposed to cadmium to survive in an adverse environment. However, excessive stress will cause tissue damage by inducing apoptosis, pyroptosis, and ferroptosis. Evidently, oxidative stress-induced autophagy plays different roles in low- or high-dose cadmium exposure-induced cell damage, either causing apoptosis, pyroptosis, and ferroptosis or inducing cell survival. Meanwhile, different cell types have different sensitivities to cadmium, which ultimately determines the fate of the cell. In this review, we provided a detailed survey of the current literature on autophagy in cadmium-induced tissue damage. A better understanding of the complex regulation of cell death by autophagy might contribute to the development of novel preventive and therapeutic strategies to treat acute and chronic cadmium toxicity.

## 1. Introduction

Cadmium-induced diseases, including kidney injury [1], osteoporosis [2], and cancer [3], caused by environmental or occupational exposure to cadmium, are rapidly becoming a global health problem because of the increased use of cadmium in industrial and agricultural applications. Unlike organic contaminants, cadmium is difficult to break down by microorganisms, and it cannot be absolutely eliminated by the body [4]. However, this chronic exposure seriously threatens human and animal health. The liver and kidneys are well-known to be extremely sensitive to acute and chronic cadmium exposure [5,6]. In addition, there was an outbreak of Itai-itai disease, which causes significant osteoporosis in patients, resulting from long-term cadmium exposure in a suburb of the City of Toyama, the capital of Toyama prefecture, Japan [7]. Epidemiological and clinical data show that cadmium can permeate the nervous system via the blood–brain barrier and cause serious neural symptoms, such as headache, vertigo, olfactory dysfunction, memory deficits, polyneuropathy, lower attention span, and Parkinsonism symptoms [8,9]. Overall, the kidney, liver, skeleton, and nervous system are the serious target organs following chronic cadmium exposure, which induce a series of organ injuries; however, the exact mechanism of autophagy in cadmium-induced organ injury or protection remains unclear. Studies have suggested that cadmium may cause various organ injuries through several mechanisms, including causing oxidative stress, DNA damage, endoplasmic reticulum stress, apoptosis, and autophagy [10,11]. In the present review, we pay particular attention to the role of oxidative stress-induced autophagy by cadmium exposure in kidney, liver, and bone damage, and neurotoxicity, and discuss the dual effects of autophagy on cadmium-induced cell death.

## 2. Autophagy

As shown in Figure 1, autophagy is a process in which cytoplasmic proteins or organelles are phagocytosed into vesicles and fused with lysosomes to form autophagic lysosomes, followed by degradation of their encapsulated contents, thus fulfilling the metabolic needs of the cell and renewing some organelles [12]. In addition, autophagy is considered a form of programmed cell death; however, its function as a regulated cell death mechanism remains controversial.

Autophagy processes are usually divided into three main categories, macroautophagy, chaperon-mediated autophagy, and microautophagy [13]. Macroautophagy (referred to as autophagy) is a highly dynamic and tightly regulated cellular event in eukaryotic cells [14]. The main process of autophagy is the formation of the autophagosome, which is mediated by a series of autophagy-related genes and key signaling pathways [15]. 

### 2.1. Autophagy Initiation

The studies have shown that Atg1 homologs Unc-51-like kinase 1 and 2 (ULK1 and ULK2) can initiate autophagy in mammalian cells [16,17], and the ULK complex consists of the ULK1/2 and other interacting proteins including ATG13 (autophagy-related 13), FIP200 (FAK family kinase-interacting protein of 200 KDa), and ATG101 [18]. ATG13 was the earliest identified member of the ULK1/2 complex, which promotes the activation of ULK1/2 [19]. FIP200 is considered a functional analog of ATG17, and participates in the assembly of ATG proteins as a scaffold protein. When cell hunger, hypoxia, and disease occur, ULK1/2, ATG13, ATG5, and LC3 could not be recruited to the membrane, and finally, autophagy was inhibited [20]. ATG101 consists of 218 amino acids and is a cytoplasmic hydrophilic protein that is widely expressed in many eukaryotic cells, but not in yeast [21]. In low-nutrition conditions, activation of the ULK1/2 complex is mainly regulated by 5′ AMP-activated protein kinase (AMPK) via phosphorylating the serine residues (467, 555, and 638) of ULK1, while simultaneously inhibiting the autophagy repressor, mammalian target of rapamycin (m-TORC1) [22,23]. In addition, AMPK can also phosphorylate the Raptor subunit of the mTORC1 complex to inhibit the activation of mTORC1 and indirectly activate ULK1 [24]. In mammals, the phosphorylation of ULK1 initiates the translocation of the VPS34 complex, which includes phosphatidylinositol 3-kinase catalytic subunit type 3 (VPS34), Beclin-1, phosphatidylinositol 3-kinase catalytic subunit type 4 (VPS15), and ATG14L, and mediates nascent phagophore formation [25]. 

### 2.2. Phagophore Elongation

Two ubiquitin-like reactions are required for the elongation of the phagophore membrane. Firstly, ATG12 binds to ATG5 through ATG7 and ATG10 (ubiquitin enzymes E1 and E2), and then the ATG5-ATG12 complex binds to ATG16 via non-covalent bonds, which forms the ATG12-ATG5-ATG16 complex (ubiquitin enzyme E3) [26]. This complex then forms the autophagosome membrane by recruiting the second complex, microtubule-associated protein light chain 3 (LC3I). LC3I covalently binds to phosphatidylethanolamine liposomes via ATG7, ATG4 (similar to E1), ATG3 (similar to E2), and the ATG12-ATG5-ATG16 complex (similar to E3) to form LC3-II [27]. LC3-II is commonly used as a marker of autophagy formation and is also an important multi-signal transduction regulator protein located on the autophagic vesicle membrane. 

### 2.3. Autophagy Maturation and Degradation

The fusion of the autophagosome and lysosome is mediated by SNARE proteins, mainly including STX17, VAMP8, and SNAP29 [28]. STX17 is located on the mature autophagosome membrane, and SNAP29 in the cytoplasm interacts with VAMP8 on the lysosome membrane to initiate membrane fusion and formation of the autophagolysosome [29]. After fusion, the autophagosome contents are degraded by a series of lysosomal hydrolases, and the degradation products are transferred to the cytosol through the lysosomal permeable membrane for reuse by the body [30]. Altogether, many signaling pathways controlled autophagy initiation, phagophore elongation, and autophagy degradation.

## 3. The Signaling Pathways of Autophagy

Many signaling pathways are known to regulate autophagy under hypoxic and low-energy conditions. mTORC1 plays a key role by phosphorylating ULK1/2 to regulate autophagy [31]. AMPK is a major player of mTORC1 and promotes autophagy under hypoxic conditions [32]. Additionally, autophagy is induced by oxidative stress, endoplasmic reticulum (ER) stress, and DNA damage to maintain cellular survival (Figure 2). 

### 3.1. Oxidative Stress

Excessive reactive oxygen species (ROS) are generated under the stimulus of several risk factors, such as environmental heavy metals, nutrition starvation, and senescence, which induce autophagy via various signaling pathways [33]. Intracellular ROS production leads to protein and DNA damage, cytotoxicity, and cell death. Nuclear factor erythroid-derived factor 2-related factor (Nrf2) is an important transcription factor that regulates oxidative stress for self-protection [34]. Nrf2 is expressed in almost all cells and tissues, and its function is influenced by Kelch-like ECH-associated protein (Keap1). In the resting state, Nrf2 is inhibited by binding to Keap1; however, during oxidative stress, Nrf2 is released by dissociation from Keap1, which activates and translocates Nrf2 to the nucleus, where it forms a heterodimer with muscular aponeurotic fibrosarcoma (Mfa), thereby activating the expression of anti-oxidant proteins and maintaining cell survival [35]. In addition, Nrf2 is activated by the excessive accumulation of p62 [36]. A large amount of p62 can bind Keap1 to form polymers or autophagosomes, which can effectively reduce the binding ability of Keap1 and Nrf2, and the formed polymers are more easily cleared through the autophagy pathway, thus further reducing the level of Keap1 and its ability to bind Nrf2. However, excessive oxidative stress has been shown to play an essential role in cadmium-induced cell damage [37,38].

### 3.2. ER Stress

The ER is an important membranous organelle for protein synthesis, folding, and secretion in eukaryotic cells. Diverse cellular stresses, such as in Ca^2+^ depletion, redox imbalance, altered protein glycosylation, or protein folding defects, cause unfolded or misfolded proteins to accumulate in the ER lumen, called ER stress. A major ER stress pathway is the unfolded protein response (UPR), which dynamically adjusts the protein folding and degradation capacity of the ER. ER stress induces an autophagic response by activating the protein kinase RNA-like ER kinase (PERK)/eukaryotic initiation factor 2α (eIF2α), inositol-requiring enzyme 1 (IRE1), and Ca^2+^ signaling [39,40]. Firstly, the activation of IRE1 kinase recruits tumor necrosis factor receptor-associated factor 2 (TRAF2) and activates C-Jun amino-terminal kinase (JNK) pathways by regulating mitogen-activated protein kinase kinase (MAPK) and apoptotic signal-regulated kinase (ASK1), respectively. JNK regulates phosphorylation of Bcl2, resulting in the dissociation of Beclin-1 from Bcl2, which regulates autophagy downstream [41]. Secondly, the activation of PERK regulates LC3 and ATG5 via the eIF2α-ATF4-CHOP signaling pathway [42]. More importantly, Ca^2+^ signals play an important role in ER stress-induced autophagy. Ca^2+^ is released from the ER to induce autophagy by activating the Ca^2+^-calmodulin-dependent protein kinase 2/β (CaMKK2/CaMKKβ)/AMP-activated protein kinase (AMPK) pathway [43,44]. Overall, ER stress benefits cellular survival by inducing autophagy; however, prolonged ER stress can induce cellular death and damage [45]. Many studies have indicated that heavy metal-induced ER stress mediates cell death by affecting autophagy-related signaling [46,47].

### 3.3. DNA Damage Response

When DNA damage occurs, cells detect it via various mechanisms and monitoring systems, resulting in systematic responses to repair the damage, called DNA damage responses (DDRs) [48]. DNA damage can be widespread in every cell, usually caused by endogenous and exogenous stimuli, such as UV light, ROS, and heavy metals [49,50]. A study has shown that DNA damage is involved in the development of many diseases, including aging and cancer [51]; however, autophagy is also essential in a range of different stimulus-induced diseases [52]. DNA damage-induced autophagy and autophagy deficiency lead to genomic instability. To date, the mechanism by which DNA damage in the nucleus triggers autophagy that occurs in the cytoplasm remains enigmatic. 

In DDR, distinct proteins recognize the damaged DNA region for DNA repair; however, the cell fate ultimately depends on the degree of DNA damage. When double-stranded DNA breaks (DSBs) occur, the protein kinases of DDR, Ataxia telangiectasia mutated (ATM) and DNA-dependent protein kinase (DNA-PK) are activated, and the Ataxia telangiectasia and Rad3-related protein (ATR) is activated after single-stranded DNA breaks (SSBs). Activation of ATM, ATR, and DNA-PK leads to the phosphorylation of many downstream proteins for DNA repair, including checkpoint kinases CHK1 and CHK2 [53]. In a study, cisplatin-induced autophagy via ATM-CHK2-dependent phosphorylation of FOXK, which is a transcription factor that has been previously associated with the suppression of *ATG* genes [54]. In addition, studies found that DNA damage induced autophagy by activating the ATM-LKB1-AMPK-TSC2 or CHK1-Beclin-1 signaling pathways [55,56]. 

p53 is a transcription factor that is involved in autophagy, apoptosis, and DNA repair [51]. Activation of p53 is strictly regulated by post-translational modifications, such as phosphorylation and ubiquitylation. When DNA damage occurs, activation of p53 contributes to the upregulation of *ATG* genes [57]. However, studies have suggested that the regulation of p53 on autophagy depends largely on its localization: cytoplasmic p53 suppresses autophagy by interfering with ULK1 complex-dependent autophagosome biogenesis, whereas nuclear p53 can increase autophagy under stress conditions [58,59]. In cancer cells, when DNA damage occurs, knockdown of p53-activated autophagy via ATM-dependent phosphorylation of PTEN [60]. In short, the regulation of autophagy by the p53 complex under stress conditions. 

In a word, oxidative stress, ER stress, and DNA damage are the main causes of cell injury. However, the toxicity of heavy metals is mainly induced by oxidative stress, which further caused ER stress, DDR and autophagy. The toxic effects of cadmium on the kidney, liver, bone, and brain are the main emphasis of this review.

## 4. The Toxicity of Cadmium toward the Kidney, Liver, Bone, and Brain

Cadmium mainly enters the human and animal bodies through the food chain. With a half-life of 10–30 years, once cadmium enters the body, it exerts extreme toxicity toward the organism. Studies have shown that cadmium exposure is closely related to fatty liver, nephropathy, neurodegenerative disease, and osteoporosis [61,62,63,64]. However, the mechanisms by which cadmium induces these diseases are still unknown. In this review, we focus on how autophagy contributes to the organ damage caused by cadmium.

### 4.1. Mechanisms of Cadmium-Induced Kidney, Liver, Bone, and Brain Toxicity

What is the proposed mechanism of cadmium-induced kidney, liver, bone, and brain injury? Based on numerous studies, oxidative stress is the critical mechanism in cadmium-induced organ injury [65]. A large amount of ROS production causes oxidative damage, which induced apoptosis, pyroptosis, ferroptosis, and autophagy. In addition to this, the mechanism of cadmium-induced skeletal toxicity is complex. The cadmium promoted osteoclast function, which is an early phenomenon of cadmium-induced osteoporosis; however, cadmium-inhibited osteoclast and osteoblast activation are the main reasons for cadmium-induced osteoporosis. In any case, oxidative stress is crucial in the potential toxicity induced by cadmium.

#### 4.1.1. The Role of Autophagy in Cadmium-Induced Kidney Damage

A study has demonstrated that the kidney is the main target after cadmium exposure, in which about 40% of cadmium is accumulated [66,67,68]. Short-term exposure to high concentrations of cadmium induces acute kidney injury (AKI), which involves many biological responses, such as autophagy and oxidative stress [69]. Autophagy deficiency causes kidney damage in many AKI models, suggesting that autophagy plays an important protective role in kidney disease [70]. Earlier studies have revealed that cadmium induces lysosomal dysfunction by blocking autophagy in AKI [71,72]. In addition, cadmium exposure causes lysosomal dysfunction and blocks the autophagy flux, and the epigenetic regulator BRD4 recovered the autophagy flux by regulating the lysosomal gene expression in the renal tubule epithelium, which inhibited cadmium-induced cytotoxicity [73]. Therefore, the above studies demonstrated that cadmium exposure destroys lysosomal function, which prevents the autophagy degradation process, ultimately leading to cell death. 

Ferroptosis is a non-apoptotic form of cell death that is dependent on intracellular accumulation of iron, resulting in an increase in toxic lipid peroxide ROS, which is implicated in many diseases. Oxidative stress can induce autophagy; however, excessive autophagy promotes ferroptosis formation [74]. By contrast, inhibition of autophagy suppresses ferroptosis in cancer cells [75]. Glutathione peroxidase 4 (GPX4) is a key index of ferroptosis, and the knockdown of GPX4 alleviated stress-induced autophagy [76]. In short, the relationships among autophagy, ferroptosis, and organelle stress are complex. However, are autophagy, ferroptosis, or autophagy-mediated ferroptosis involved in cadmium-induced kidney damage? Recently, a study found that cadmium-induced ferroptosis in kidney and renal tubular epithelial cells, and the inhibition of autophagy mitigated cadmium-induced ferroptosis, which suggested that ferritinophagy disrupted iron regulation and contributed to cadmium-induced ferroptosis [77]. Therefore, ferroptosis is involved in cadmium-induced kidney toxicity via regulating ferritinophagy. Meanwhile, oxidative stress-induced autophagy may be a key point in ferroptosis formation.

Apoptosis is critical for cadmium-induced kidney damage. Studies have suggested that autophagy has protective effects against cadmium-induced apoptosis and necrotic cell death in kidney injury [78,79]. A study found that cadmium exposure (at concentrations of 5, 10, 20, 30, and 40 μM) for 30 h increased the accumulation of LC3II and p62, which impaired p62-mediated autophagy or resulted in a low-autophagic flux, leading to apoptosis [80]. In that study, the authors demonstrated the effect of Sirt6 on autophagy degradation: the knockdown of p62 promoted Sirt6 nuclear transposition and inhibited kidney mesangial cell apoptosis. p62 is a ubiquitin-binding protein, which accumulates when autophagy is inhibited. Another study found that cadmium exposure (at concentrations of 0.5, 2, 10, and 20 μM) promoted autophagy development in rat renal mesangial cells for 24 h, in which the accumulation of p62 was significantly reduced and cell viability was good; however, after autophagy was compromised by knockout of *Atg16*, cell viability decreased and cadmium increased the extent of apoptosis [81]. Therefore, these differences are largely caused by different cadmium exposure concentrations and times. Consistently, a study showed that the renal proximal tubule (PT) is also a key target of cadmium toxicity. The authors chose the PT cell line, NRK-52E, as a kidney toxicity model, and after cadmium exposure (at concentrations of 5–10 μM) for 1 h, the expression of LC3-II increased, and the accumulation of p62 decreased, accompanied by ER-stress; however, the cells did not show apoptosis [82]. Cadmium exposure for more than 3 h resulted in p62/LC3-II accumulation. After exposure for 24 h, cadmium (5–25 μM) activated Bax/Bcl-2/PARP-1 signaling, which promoted apoptosis. These results indicated that low cadmium exposure rapidly activated autophagy by inducing ER stress to counteract the damage; high cadmium exposure disrupted the autophagic flux and lysosomal stability, finally resulting in apoptosis. In conclusion, autophagy induced by exposure to cadmium at different concentrations and for different periods is closely related to apoptosis. 

Pyroptosis, also known as cell inflammatory necrosis, is a kind of programmed cell death. It causes cell swelling until the cell membrane ruptures, resulting in the release of cell contents and the activation of a strong inflammatory reaction. Pyroptosis is an important natural immune response of the body, playing a vital role in fighting infection. Pyroptosis is programmed cell necrosis mediated by Gasdermin [83]. Interestingly, a study has indicated that pyroptosis and autophagy interact [84]. Zhang et al. found that cadmium and molybdenum co-exposure induced duck renal tubular epithelial cell pyroptosis and autophagy [85,86]. However, the interaction between cadmium-induced renal autophagy and pyroptosis remains unclear. In Zhang et al.’s study, pyroptosis and autophagy co-induced by molybdenum and 4 μM cadmium for 12 h were interrelated in duck renal tubular epithelial cells, and inhibiting pyroptosis might attenuate autophagy, but suppressing autophagy might promote pyroptosis [86]. Unfortunately, the relationship between cadmium-induced pyroptosis and autophagy is poorly understood in renal injury. Therefore, it is important to determine the mechanism of autophagy-mediated pyroptosis in cadmium-induced renal injury. 

In a word, apoptosis is the main reason for cadmium-induced kidney injury; however, the toxicity mechanism of cadmium-induced pyroptosis and ferroptosis in kidney damage remains unknown.

#### 4.1.2. The Role of Autophagy in Cadmium-Induced Liver Damage

Except that the kidney is the main target organ after cadmium enters the body, 30% is accumulated in the liver [87]. Therefore, the liver is also a main target organ of cadmium toxicity. However, very limited information has been documented regarding the role of autophagy in cadmium-induced liver damage.

Recently, Wang et al. established a model of cadmium toxicity in pigs. Pigs were fed a basic diet supplemented with cadmium (15 ± 0.242 mg/kg) for 30 days because of pigs’ anatomical and physiological similarity to humans [88]. In that study, they aimed to explore the mechanism of cadmium-induced liver toxicity in pigs. The results suggested that, in response to cadmium, apoptosis was significant in liver tissue, ROS levels increased remarkably, the accumulation of p62/LC3II increased significantly, and AMPK/PPAR-γ/NF-kB signaling was activated. Notably, cadmium exposure induced oxidative stress and impaired autophagic degradation, which caused apoptosis. In addition, studies have reported that cadmium exposure is associated with hepatic necroinflammation and non-alcoholic fatty liver disease (NAFLD) [89,90]. The feature of NAFLD is lipid accumulation in hepatocytes; however, autophagy can degrade lipids in the cytoplasm. Interestingly, Rosales et al. reported that cadmium (25 nM) upregulated the expression of anti-apoptotic Bcl2, and increased the MT-II content in cholesterol overload-induced hepatocytes; meanwhile, cadmium exposure further increased the lipid content in hepatocytes from cholesterol-fed mice and impaired autophagy by reducing the expression of LC3II via activating m-TOR signaling [91]. By contrast, the opposite effect was observed in hepatocytes from control standard chow-fed mice. In short, cadmium exposure exacerbates hyperlipidemia in cholesterol-overloaded hepatocytes via autophagy dysregulation. In the study, they found that hepatocytes from high cholesterol (HC) fed mice were not sensitive to cadmium compared with control hepatocytes from standard chow-fed mice, which is the main reason why MT-II could scavenge the heavy metal [91]. Even so, cadmium exposure impaired autophagy, which inhibited triglyceride clearance and caused serious liver disease. Zou Hui et al. [92] also reported that cadmium exposure (2.5, 5, 10 μM) for 6 h promoted rat liver cell apoptosis, and the expression of autophagy-related proteins significantly increased. However, the autophagy inducer, Rapamycin, prevented the cadmium (2.5 μM)-induced cytotoxicity in rat liver cells. Thus, autophagy, as a protective mechanism, plays an important role in cadmium-induced liver disease. 

Autophagy can be used as a stress adaptation and protection mechanism in some cases, but over-activated autophagy will induce cell death under severe oxidative stress [93]. Zhang et al. found that cadmium exposure (5 μM) for 24 h induced apoptosis and increased ROS production, thereby activating autophagy in chicken hepatocytes [93]. However, selenium (Se) induced the transcription of *NRF2*, which inhibited the production of a large amount of ROS, induced by cadmium, which prevented autophagy and apoptosis [94]. Gong et al. also found a similar phenomenon, trehalose prevented cadmium-induced hepatotoxicity by blocking the Nrf2 pathway, thereby restoring autophagy and inhibiting apoptosis [95]. Zhou et al. reported that puerarin prevents cadmium-induced hepatic cell damage by suppressing apoptosis and restoring autophagic flux [96]. According to that report, cadmium exposure disrupted autophagy flux and then induced apoptosis, and a range of antioxidants play an essential role in cadmium toxicology. Therefore, the inhibition of oxidative stress may be an effective means of preventing cadmium liver toxicity.

Pyroptosis is inflammation-mediated programmed cell death, and Pengcheng et al. found that cadmium exposure induced oxidative stress and activated NLRP3/caspase-1-dependent pyroptosis in chicken liver cells [97]. However, selenium (Se) significantly decreased cadmium-induced oxidative stress, and prevented chicken liver pyroptosis. In that study, the author did not report the change in autophagy level after cadmium application. A study has reported that oxidative stress can cause autophagy or autophagy-mediated cell death [98]. Therefore, Se might inhibit oxidative stress and recover autophagy flux, finally preventing the liver cell pyroptosis. In another study, Cao et al. [99] reported that cadmium and molybdenum induced pyroptosis and apoptosis via PTEN/PI3K/AKT signaling in duck liver tissue; however, the relationship between apoptosis and pyroptosis was not reported, and how autophagy regulated pyroptosis and apoptosis was also not explained. That study suggested that the most important function of PTEN is to block the activation of the PI3K-AKT-m-TOR signaling pathway via its lipid phosphatase activity [99]. In that study, the results indicated that cadmium induced PTEN activation and inhibited PI3K/AKT signaling, allowing us to speculate that cadmium might promote autophagy development, but prevents autophagy degradation, causing pyroptosis and apoptosis. In addition, pyroptosis and apoptosis are likely to occur simultaneously in duck liver tissue; however, pyroptosis and apoptosis might be affected by cadmium exposure concentrations and time in vitro. In summary, it is important to determine the relationship between autophagy and pyroptosis induced by cadmium. However, cadmium exposure-induced ferropotosis in liver injury has not yet been reported; therefore, the relationship between cadmium-induced autophagy and ferropotosis should be the focus of future research in liver diseases. 

#### 4.1.3. The Role of Autophagy in Cadmium-Induced Bone Damage

Cadmium exposure has long been associated with osteoporosis. The disease itai-itai is mainly caused by long-term consumption of cadmium-contaminated rice in Japan, which results in fractures, mainly in postmenopausal women, together with a form of osteoporosis and kidney dysfunction [100]. The basic mechanism of bone mass and bone strength decline in the body comprises age-related changes. However, osteocyte death mainly occurs with the reduction in bone strength after aging, as compared with osteoblasts and osteoclasts, which have a longer survival period, and the possible mechanism leading to osteocyte death is the failure of autophagy [101]. Osteoporosis is likely to be caused by the failure of autophagy such that osteocytes cannot survive, which would accelerate the rate of age-related bone cell death [102]. In addition, osteoclasts are bone resorption cells, and excessive activity of osteoclasts leads to bone loss and osteoporosis; hence, inhibiting osteoclast autophagy is clearly beneficial to maintaining bone mass [103]. Osteoblasts are bone-forming cells, and decreased autophagy of osteoblasts or pre-osteoblasts can result in reduced bone formation and bone quality, ultimately leading to osteoporosis [104]. However, the molecular mechanism of autophagy in cadmium-induced bone injury remains unclear. This review mainly focuses on the role of autophagy in cadmium-induced bone cell injury.

Liu et al. found that cadmium exposure (1, 2, and 5 μM) for 3 h induced apoptosis in rat primary osteoblasts and initiated autophagy development [105]. However, cadmium-induced osteoblast apoptosis was suppressed by activating autophagy with rapamycin. Therefore, osteoblast apoptosis was called autophagy defect death, which was not affected by autophagy degradation. Thus, the activation of autophagy is beneficial to the survival of osteoblasts. Ran et al. also found that cadmium exposure (2, 5, and 10 μM) for 2 h activated autophagy via the AMPK/m-TOR/ULK1 pathway in rat osteoblasts, in which the expression of p62 was obviously reduced, indicating that autophagy flow was smooth [106]. The activation of autophagy avoids osteoblast apoptosis, which inhibits the formation of osteoporosis. However, Zheng et al. found that cadmium (5 and 10 μM) exposure for 12 h induced DNA damage and apoptosis in rat osteoblasts [107]. The authors of that study did not specify the autophagy level, but they did mention that cadmium-induced osteoblast apoptosis might function by preventing autophagy flux, which inhibited the degradation progress. In summary, autophagy is involved in the process of osteoblast response to cadmium-induced oxidative stress and toxicity. 

A study has suggested that autophagy contributes to osteoclast differentiation from the pre-osteoclast to the mature osteoclast [108]. In addition, the activation of autophagy’s protective effect against osteoclast apoptosis during RANKL-induced osteoclastogenesis has been reported [109]. Another study found that high levels of oxidative stress promoted autophagy activation, which contributed to osteoclast survival [110]. However, the mechanism of autophagy in cadmium-induced changes in osteoclast function remains unclear. Studies reported that low-dose–short-term (12 h) cadmium (100 nM) exposure increased autophagic degradation in osteoclasts, which promoted osteoclast differentiation and cellular viability [111]. By contrast, long-term cadmium (100 nM) exposure prevented autophagy flux, which inhibited osteoclast differentiation and promoted apoptosis, ultimately causing osteoporosis [111]. Chronic and acute low-dose cadmium exposure caused bone loss, which is the main reason why cadmium delays osteoclast apoptosis, which suggested that cadmium has a transient antiapoptotic effect and induces bone loss by increasing osteoclast lifespan [112]. However, the authors did not refer to the effect of autophagy, which might play an important role in the cadmium-induced delay of osteoclast apoptosis. According to the above reports, activation of autophagy promoted osteoclast survival and accelerated bone loss, which explains the bone loss induced by low-dose cadmium exposure. In addition, long-term exposure to high-dose cadmium induces osteoporosis, often because of the death of osteoblasts and osteoclasts. In this case, autophagy induces cell death. In brief, the duality of autophagy plays an important role in cadmium-induced bone metabolism disorders; meanwhile, it is clear that osteoblast apoptosis is a key point in cadmium-induced osteoporosis.

#### 4.1.4. The Role of Autophagy in Cadmium-Induced Neurotoxicity

In terms of neurotoxicity, cadmium exposure has been demonstrated as a possible pathogen for many neurodegenerative diseases, including Parkinson’s disease (PD), Alzheimer’s disease (AD), and amyotrophic lateral sclerosis (ALS) [113,114,115]. However, blood–brain barrier (BBB) dysfunction is associated with neurodegenerative disorders [116]. A study has reported that cadmium can enter the cells by destroying the BBB, when cadmium is present in the bloodstream [117], which causes neurotoxicity. In this review, we will focus on the role of autophagy in cadmium-induced neurotoxicity.

Cell viability may decrease when the misfolded proteins are not efficiently degraded. Wang et al. [118] found that cadmium exposure (5, 10, and 20 μM) for 24 h induced the ER stress by increasing chaperone GRP78 expression, which caused autophagy activation. They also found that blocking autophagy by knockdown with Atg5 further increased cadmium-induced PC12 cell senescence. Indeed, ER stress can promptly respond to misfolded and unfolded proteins. In this process, autophagy is the key process for removing excess proteins. By contrast, excessive unfolded protein accumulation might cause neurodegenerative disorders. In addition, ROS overproduction might be associated with mitochondrial membrane potential breakdown [119]. These interesting data were confirmed in different neuron-like cells. According to Yan et al., cadmium exposure (5, 10, and 20 μM) induced cortical neuron apoptosis, which is mediated by oxidative stress and mitochondrial dysfunction [120]. Wen et al. [121,122] further found that puerarin protects against cadmium-induced (10 μM) neurotoxicity by alleviating mitochondrial damage, and inhibiting mitophagy-mediated mitochondrial mass decrease. In vivo, their results demonstrated that puerarin inhibited cadmium-induced neurotoxicity by alleviating oxidative stress and apoptosis in rat cerebral cortical neurons. Tang et al. found that trehalose alleviates cadmium-induced brain damage by ameliorating oxidative stress, autophagy inhibition, and apoptosis [123]. Therefore, it is important to develop new antioxidant technologies to prevent cadmium-induced neurotoxicity. 

Li et al. designed a chronic cadmium-induced neurotoxicity experiment in mice that lasted 30 months [124]. They found that the toxicity of chronic cadmium exposure was associated with the transition from autophagy to apoptosis, and the autophagy-apoptosis switch was cadmium dose-dependent, with a threshold of [Cd^2+^] 0.04 mg/L. They thought that hippocampus-dependent learning and memory damage was mainly caused by neuronal apoptosis. Wang et al. reported that cadmium exposure (2.5, 5, 10, and 20 μM) for 12 h induced a high level of autophagy in PC12 cells [125]. They also found that induction of autophagy by cadmium might exert a cytoprotective role. However, Xu et al. found that cadmium exposure (2.5, 5, 10, and 20 μM) for 24 h induced the expansion of autophagosomes in PC12 cells and primary murine neurons [126]. 3-MA (an autophagy inhibitor) attenuated cadmium-induced expansion of autophagosomes and apoptosis. In another study, the authors found that cadmium results in the accumulation of autophagosome-dependent apoptosis through activating Akt-impaired autophagic flux in neuronal cells [127]. Pi et al. reported a new mechanism in which cadmium exposure (12.5, 25, 50 μM) for 24 h in Neuro-2a cells induced lysosomal membrane permeabilization and eventually triggered caspase-dependent apoptosis [128]. The authors believed that TFE3 is significantly important in cadmium-induced neurotoxicity by maintaining the lysosomal-mitochondrial axis; however, the protective effect of TFE3 is not dependent on the restoration of autophagic flux. According to the above reports, it appears that autophagy plays a different role in cadmium-induced neurotoxicity according to the cadmium concentration. 

Aside from apoptosis, pyroptosis and ferroptosis also play an essential role in cadmium-induced neurotoxicity. A study found that NLRP3 inflammasome-mediated pyroptosis is involved in cadmium-induced neuroinflammation through the IL-1β/IkB-α-NF-κB-NLRP3 feedback loop in swine [129]. In duck brains, a study found that cadmium and molybdenum co-induced pyroptosis by inhibiting the Nrf2-mediated antioxidant defense response [130]. In a recent report, cadmium (2 and 4 μM) induced ferroptosis and apoptosis in PC12 cells by regulating the miR-34a-5p/Sirt1 axis [131]. In a cadmium induction experiment in swine, cadmium exposure induced ferroptosis pathways by activating oxidative stress, eventually causing cerebrum and cerebellum damage [132]. However, ferroptosis and pyroptosis have been rarely reported in cadmium-induced neurotoxicity, and the relationship between autophagy and ferroptosis and pyroptosis has not been reported in cadmium-induced neurotoxicity. Therefore, it is important to determine the role of autophagy in cadmium-induced ferroptosis and pyroptosis in future studies.

## 5. Conclusions and Future Perspectives

Based on the current state of knowledge and the literature, it is essential to explore the relationship between autophagy and apoptosis, pyroptosis, and ferroptosis in cadmium-induced organ damage in further detail. According to current research, low-dose cadmium exposure promotes the survival and proliferation of cells, especially in bone cells, which might be the main reason for cadmium-increased osteoclast differentiation and osteoporosis. Alternatively, it might act as a carcinogen, by further inducing the proliferation of primary cancer cells and aggravating cancer development, especially in liver cancer. In the above process, autophagy plays a protective mechanism. In addition, high-dose cadmium exposure enhances cell death by a variety of mechanisms, leading to organ damage, such as neurodegenerative disease, kidney damage, liver damage, and osteoporosis. In this case, autophagy plays two roles in cadmium-induced cell death: on the one hand, cadmium exposure prevents autophagy flux and destroys lysosome function, which inhibits autophagolysosome degradation, finally causing cell death. However, recovery of autophagy flux by intervention using medicine, such as resveratrol, puerarin, and taurine, can effectively alleviate cadmium-induced cell death. On the other hand, hyperactivation of autophagy can also induce cell death, which can be prevented using the autophagy inhibitor, 3-MA. In either case, the outcome of cell exposure to cadmium depends not only on the duration and concentration of exposure, but also on other factors inherent in the cell itself. For example, low-dose cadmium exposure activates autophagy, which plays a protective role; on the contrary, high-dose cadmium exposure prevents autophagy, which promotes cell injury. On the other hand, poultry osteoblasts are more sensitive to cadmium than mouse osteoblasts, and primary cells are more sensitive than cell lines. In conclusion, the mechanism of autophagy in cadmium-induced cell death is complex and requires further study. 

In this review, we focused on the relationship between cadmium-induced apoptosis and autophagy; however, the exact mechanism needs to be further clarified. Oxidative stress, ER stress, and DNA damage can activate autophagy, leading to cell survival or apoptosis. However, cell apoptosis depends on the extent of the stress. Interestingly, among the many cadmium-induced cell death methods, apoptosis is the most reported. However, how cells are transformed to undergo cadmium-induced apoptosis, pyroptosis, and ferroptosis remains unclear, and it is also unclear whether autophagy regulates pyroptosis and ferroptosis. Surprisingly, cadmium-induced pyroptosis and ferroptosis have not been reported in bone cells and hepatocytes. Animal liver is extremely high in iron; therefore, cadmium-induced liver injury is likely to be associated with ferroptosis. Its increased lipid peroxidation and ROS cause the induction of autophagy. Overall, the relationship between autophagy and ferroptosis will be the focus of cadmium toxicity-related research in the future. Pyroptosis is different from ferroptosis, comprising inflammation-mediated programmed death, and many studies have demonstrated that cadmium activates the NF-kB signaling and promotes the activation of inflammasomes. A study also reported that suppressing inflammation and downstream NF-kB activated pyroptosis pathways, which inhibited postmenopausal osteoporosis [133]. Therefore, pyroptosis might be involved in cadmium-induced osteoporosis, which provides a novel direction for the treatment of osteoporosis. Although, cadmium-induced kidney damage and nerve injury, the evidence is not convincing; therefore, it would be more accurate to use pyroptosis-like and ferroptosis-like. Therefore, cadmium-induced pyroptosis and ferroptosis need to be further verified in different primary cells and cells line. In addition, whether the different levels of cadmium-induced different cell death occur simultaneously or are transformed mutually remains to be determined. Finally, there is an urgent need to discover how autophagy-related genes regulate different modes of cell death.

Hopefully, a detailed understanding of the mechanism of autophagy-regulated cell death and its role in cell fate decisions might ultimately contribute to the development of novel strategies for the prevention and therapy of acute and chronic cadmium-induced neurotoxicity, hepatotoxicity, nephrotoxicity, and bone toxicity (Figure 3).

## Figures and Tables

**Figure 1 ijms-23-13491-f001:**
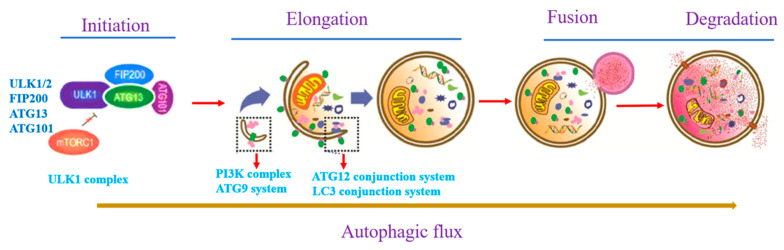
Patterns of autophagic flux, which is composed of autophagy initiation, elongation, and degradation, and every process is regulated by many autophagy-related genes, such as ATG13, ATG12, and LC3.

**Figure 2 ijms-23-13491-f002:**
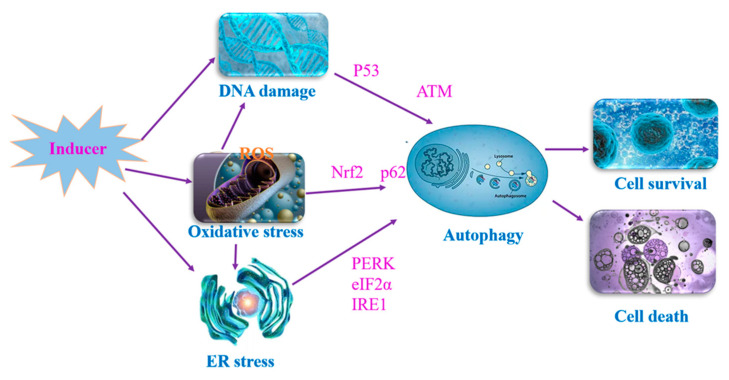
Molecular pathways involved in the regulation of autophagy. Oxidative stress induced DNA damage and ER stress, which caused autophagy development. Autophagy has a dual nature, which can not only save cell survival, but also promote cell death.

**Figure 3 ijms-23-13491-f003:**
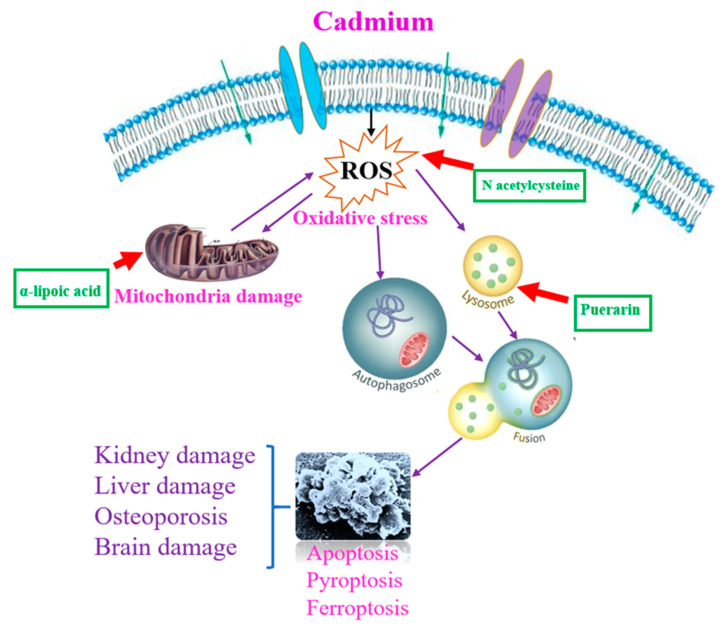
Possible mechanism targets of autophagy-regulated cell death. N-acetylcysteine, puerarin and α-lipoic acid may be an effective protective agent for the prevention and therapy of acute and chronic cadmium-induced neurotoxicity, hepatotoxicity, nephrotoxicity, and bone toxicity.

## Data Availability

Not applicable.

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
