# Peer review of "The Effect of Oxidative Stress-Induced Autophagy by Cadmium Exposure in Kidney, Liver, and Bone Damage, and Neurotoxicity"

_ijms, 2022, doi:10.3390/ijms232113491_

Round 1

Reviewer 1 Report (New Reviewer)

In their paper entitled: "The effect of oxidative stress induced autophagy by cadmium exposure in kidney, liver, and bone damage, and neurotoxicity" the authors bring current information from the literature on the mechanism of autophagy. Particular attention was paid to cadmium exposure, and the influence was described at different levels and in different animal models. The review is well elaborated, with appropriate references provided.

The only comment relates to a typo on page 11, line 504: its increased lipid... should start with a capital letter.

Author Response

Dear editor, 

      we have modified the question.

Yonggang Ma

Reviewer 2 Report (New Reviewer)

Dear authors thank you for article, I enjoyed reading it! I think that this paper deserve to be published in IJMS.

Only just a comment, in the text highlighted in red the references are not consequential to the previous ones. Please revise them. 

Author Response

Dear editer, 

I have modified the question.

Yonggang Ma

This manuscript is a resubmission of an earlier submission. The following is a list of the peer review reports and author responses from that submission.

Round 1

Reviewer 1 Report

This is an interesting and timely review that attempts to integrate information about how Cd induces toxicity in several target tissues. While the language is reasonably good for the most part, some editing for proper grammar is needed. The figures are very good and encapsulate key points very nicely. With that said, however, there are some issues with overall presentation and rationale. There are several statements made by the authors that imply that there is no information available about how Cd acts in different tissues. This is misleading and incorrect. The other major issue concerns the organization and focus of the paper. Different sections need better introduction with clear transition statements.

Specific Comments:

1.         Lines 16-17: Such a broad statement is empty and misleading; there is much known about how Cd acts.

2.         Line 35: Illogical statement; Cd is eliminated but very slowly.

3.         The Introduction needs to provide a clearer focus for the review. The following two sections discuss autophagy and there is no mention of Cd. It would be helpful to note that the basic processes underlying autophagy will be summarize before considering their role in Cd toxicity.

4.         Lines 45-46: Awkward and unclear.

5.         Line 75: The sentence beginning with "When" is awkward and unclear.

6.         Lines 132, 133: I believe that "P62" should be written with a small "p" as "p62."

7.         Lines 181-182: This last sentence of the paragraph is odd to me as it says nothing about Cd, which is supposed to be the main focus here??

8.         Lines 183-192: As with p62, "P53" should be written as "p53."

9.         Line 195: What is “serious” toxicity? This is the wrong descriptor.

10.      Line 201: Surely there are quite a few published studies that have shown the kidneys to be a main target for Cd, not just the one study cited here.

11.      Line 241: There is no cell line simply called "NRK." I presume you are referring to NRK-52E cells.

12.      Lines 269-270: Again, such a broad statement is incorrect and misleading. Surely some information about the mechanism of Cd-induced liver damage is known.

13.      Lines 464-465: This is confusing. What do resveratrol, pueararin, and taurine have to do with traditional Chinese medicine?

14.      In the conclusions section (section 5), there are some comments related to effects of Cd at low vs. high doses. This topic seems largely ignored in the review and needs to be better highlighted.

Author Response

Dear Editors and Reviewer:

    Thank you for your letter and for the reviewers’ comments concerning our manuscript entitled “The effect of autophagy-oxidative stress in cadmium-induced kidney, liver, and bone damage, and neurotoxicity” ijms-1869921. Those comments are all valuable and very helpful for revising and improving our paper. We have studied comments carefully and have made correction which we hope meet with approval. The main corrections in the paper and the responds to the reviewer’s comments are as flowing:

Reviewer #1:

Specific Comments:

Question 1: Lines 16-17: Such a broad statement is empty and misleading; there is much known about how Cd acts.

Response: we have modified this sentence. However, the mechanism by which cadmium induces autophagy in these diseases remains unclear.

Question 2:  Line 35: Illogical statement; Cd is eliminated but very slowly.

Response: we have deleted “because of its half-life of 10–30 years”.

Question 3: The Introduction needs to provide a clearer focus for the review. The following two sections discuss autophagy and there is no mention of Cd. It would be helpful to note that the basic processes underlying autophagy will be summarize before considering their role in Cd toxicity.

Response: we have provided a clear focus for the review. “In the present review, we focus specifically on the role of oxidative stress induces autophagy by cadmium exposure in kidney, liver, and bone damage, and neurotoxicity, and discuss the dual effects of autophagy on cadmium-induced cells death.”

Question 4: Lines 45-46: Awkward and unclear.

Response: we have modified this sentence.In all, the kidney, liver, skeleton, and nervous system are the critical target organs following chronic cadmium exposure, which induce a serious of organs injury; however, the exact mechanism of autophagy in cadmium‑induce organs injury or protection remains elusive”.

Question 5: Line 75: The sentence beginning with "When" is awkward and unclear.

Response: we have modified, “The ULK1/2, ATG13, ATG5 and LC3 could not be recruited to the membrane when hunger, hypoxia and disease”.

Question 6: Lines 132, 133: I believe that "P62" should be written with a small "p" as "p62."

Response: we have modified.

Question 7: Lines 181-182: This last sentence of the paragraph is odd to me as it says nothing about Cd, which is supposed to be the main focus here??

Response: we have deleted this sentence.

Question 8: Lines 183-192: As with p62, "P53" should be written as "p53."

Response: we have modified.

Question 9: Line 195: What is “serious” toxicity? This is the wrong descriptor.

Response: we have modified. “It exerts extremely toxicity toward the organism.”

Question 10: Line 201: Surely there are quite a few published studies that have shown the kidneys to be a main target for Cd, not just the one study cited here.

Response: we have added.

Question 11: Line 241: There is no cell line simply called "NRK." I presume you are referring to NRK-52E cells.

Response: we have modified.

Question 12: Lines 269-270: Again, such a broad statement is incorrect and misleading. Surely some information about the mechanism of Cd-induced liver damage is known.

Response: we have modified. “However, the role of autophagy in cadmium-induced liver damage has not been determined.”

Question 13:  Lines 464-465: This is confusing. What do resveratrol, pueararin, and taurine have to do with traditional Chinese medicine?

Response: we have deleted “traditional Chinese”.

Question 14: In the conclusions section (section 5), there are some comments related to effects of Cd at low vs. high doses. This topic seems largely ignored in the review and needs to be better highlighted.

Response: we have modified. Lines 476-478.

Yours sincerely,

Dr. Liu

College of Veterinary Medicine, Yangzhou University

12 Wenhui East Road

Yangzhou 225009, P R China

Tel: +86-514-87979042

Fax: +86-514-87972218

E-mail: liuzongping@yzu.edu.cn

Reviewer 2 Report

ijms-1869921

Type Review

TITLE: The effect of autophagy-oxidative stress in cadmium-induced kidney, liver, and bone damage, and neurotoxicity

Authors:Yonggang Ma , Qunchao Su , Chengguang Yue , Hui Zou , Jiaqiao Zhu , Hongyan Zhao , Ruilong Song, Zongping Liu *

Ma et al provide a review about mechanisms implicated in autophagy in tissues damaged by exposure to cadmium with the aim of smoothing the path for new therapeutic strategies for treating acute and chronic cadmium toxicity. They describe first the autophagy, the underlying mechanisms, and the signalling pathways that activate the process, then they analyse the effects of cadmium exposure and the related tissue damage in kidney, liver, bone, and brain. Literature is very recent: more than half of the references is among 2017-2022 years. The paper is suitable for the special issue: Cellular and Molecular Mechanisms of Heavy Metal Toxicity: From Death to Immortality. However, before acceptance for publication, some changes are required to improve the manuscript.

The title should be more comprehensive: i.e. The effect of oxidative stress induced autophagy by cadmium exposure in kidney, liver, and bone damage, and neurotoxicity.

There are several abbreviations and, to make the manuscript more comprehensive, a list should be added.

Paragraph page 8, lanes 339-344, should be referenced.

In the paragraph of the conclusions, especially in the last sentences, words such as pyroptosis, ferroptosis and autophagy are continuously repeated, authors should make the text linguistically more fluid.

Author Response

Dear Editors and Reviewer:

    Thank you for your letter and for the reviewers’ comments concerning our manuscript entitled “The effect of autophagy-oxidative stress in cadmium-induced kidney, liver, and bone damage, and neurotoxicity” ijms-1869921. Those comments are all valuable and very helpful for revising and improving our paper. We have studied comments carefully and have made correction which we hope meet with approval. The main corrections in the paper and the responds to the reviewer’s comments are as flowing:

Reviewer #2:

Question 1: The title should be more comprehensive: i.e. The effect of oxidative stress induced autophagy by cadmium exposure in kidney, liver, and bone damage, and neurotoxicity.

Response: the title has modified.

Question 2: There are several abbreviations and, to make the manuscript more comprehensive, a list should be added.

Response: we have added.

Question 3: Paragraph page 8, lanes 339-344, should be referenced.

Response: we have added.

Question 4: In the paragraph of the conclusions, especially in the last sentences, words such as pyroptosis, ferroptosis and autophagy are continuously repeated, authors should make the text linguistically more fluid.

Response: we have deleted part words.

Yours sincerely,

Dr. Liu

College of Veterinary Medicine, Yangzhou University

12 Wenhui East Road

Yangzhou 225009, P R China

Tel: +86-514-87979042

Fax: +86-514-87972218

E-mail: liuzongping@yzu.edu.cn

Round 2

Reviewer 1 Report

While the authors have corrected or removed most of the identified misstatements from the previous version, English language and grammar remain poor. In fact, most of the newly added text has grammatical errors. Professional editing is needed before this can be accepted.

Other comments:

1.         Line 45: Series? “organs” should be singular.

2.         Line 77: Hunger?

3.         Line 109: “pathway” should be plural.

4.         Lines 193-196: This newly added sentence is awkward and needs rewriting.

5.         Line 199: Both the form and word itself (“extremely”) are incorrect.

6.         Line 203: needs grammar correction.

7.         Line 276: change “live” to “liver.”

8.         Line 344: The newly added text is grammatically incorrect.

Reviewer 2 Report

Authors addressed my comments, and, in my opinion, the revised manuscript is now suitable for publication.